# Intravesical Recurrence after Radical Nephroureterectomy in Patients with Upper Tract Urothelial Carcinoma Is Associated with Flexible Diagnostic Ureteroscopy, but Not with Rigid Diagnostic Ureteroscopy

**DOI:** 10.3390/cancers14225629

**Published:** 2022-11-16

**Authors:** Jee Soo Ha, Jinhyung Jeon, Jong Cheol Ko, Hye Sun Lee, Juyeon Yang, Daeho Kim, June Seok Kim, Won Sik Ham, Young Deuk Choi, Kang Su Cho

**Affiliations:** 1Department of Urology, Prostate Cancer Center, Urological Science Institute, Gangnam Severance Hospital, Yonsei University College of Medicine, Seoul 06273, Republic of Korea; 2Biostatistics Collaboration Unit, Yonsei University College of Medicine, Seoul 03722, Republic of Korea; 3Department of Urology, Urological Science Institute, Severance Hospital, Yonsei University College of Medicine, Seoul 03722, Republic of Korea; 4Center of Evidence Based Medicine, Institute of Convergence Science, Yonsei University, Seoul 03722, Republic of Korea

**Keywords:** ureteral neoplasms, urinary bladder neoplasms, ureteroscopy

## Abstract

**Simple Summary:**

Diagnostic ureteroscopy (URS) before radical nephroureterectomy is a risk factor for intravesical recurrence in patients with upper tract urothelial carcinoma. Although flexible URS requires a higher-pressure inflow of irrigation fluid than that of rigid URS, previous studies have not considered mechanical differences in relation to the type of URS. In this manuscript, we assessed the impact of diagnostic URS on intravesical recurrence following radical nephroureterectomy for upper tract urothelial carcinoma according to the type of URS.

**Abstract:**

(1) Background: We assessed the impact of diagnostic ureteroscopy (URS) on intravesical recurrence (IVR) following radical nephroureterectomy (RNU) for upper tract urothelial carcinoma according to the type of URS. (2) Methods: Data on 491 consecutive patients who underwent RNU at two institutions between 2016 and 2019 were retrospectively reviewed. The study population was classified according to the type of URS performed before RNU as follows: non-URS, rigid URS, and flexible URS. The study outcome was IVR occurring within 1 year of RNU. Univariable and multivariable Cox proportional hazards models were used to estimate the risk of IVR. (3) Results: Altogether, 396 patients were included for analysis. Rigid and flexible URS were performed in 178 (45%) and 111 (28%) patients, respectively, while 107 (27%) patients did not undergo URS. IVR was identified in 99 (25%) patients. Multivariable Cox regression analysis revealed that the flexible URS group was significantly associated with increased IVR, compared to the non-URS group (HR = 1.807, *p* = 0.0416). No significant difference in IVR was observed between the non-URS and rigid URS groups (HR = 1.301, *p* = 0.3388). (4) Conclusions: In patients with UTUC undergoing RNU, rigid URS may not increase the risk of IVR, whereas flexible URS appears to be associated with a higher risk of IVR.

## 1. Introduction

Upper urinary tract urothelial carcinoma (UTUC) is a rare malignancy, accounting for 5–10% of all urothelial carcinomas [1,2]. According to a recent meta-analysis, computed tomography (CT)-urography shows a 92% sensitivity and 95% accuracy [3] in the diagnoses of UTUC, and diagnostic ureteroscopy (URS) is recommended only when imaging studies and cytology are insufficient for diagnosis [4]. However, although diagnostic URS can reduce the misdiagnosis rate to 2.1% [5], it is reportedly a risk factor for intravesical recurrence (IVR) after radical nephroureterectomy (RNU) [6,7,8].

The rate of IVR after RNU for UTUC is reportedly 22–47% [9,10]. Downward seeding, as well as field cancerization, is understood to be a mechanism of IVR [11,12,13]. Shigeta et al. recently revealed that IVR tumors after RNU exhibit molecular characteristics similar to those in UTUC via a comparative study of UTUC origin and primary bladder tumor [14]. Seisen et al. reported several predictors of IVR in their systematic review, including patient-specific (male sex, previous bladder cancer, preoperative chronic kidney disease), tumor-specific (positive preoperative urinary cytology, ureteral location, multifocality, invasive pathological T stage, necrosis), and treatment-specific (laparoscopic RNU, extravesical bladder cuff removal, positive surgical margin) predictors [15]. Other studies suggested that diagnostic URS prior to RNU is a risk factor for higher IVR, although this remains controversial [6,7,16]. Two recent studies, however, demonstrated that URS prior to RNU increases the risk of IVR only when combined with endoscopic biopsy, albeit with no concurrent impact on the other long-term survival outcomes [8,17].

Over the last decade, flexible URS has become a popular modality in the exploration of the entire upper urinary tract, offering improved diagnostic accuracy due to advances in endourologic technologies. Several novel optical diagnostic techniques, such as narrow band imaging, photodynamic diagnosis, and optical coherence tomography, have been introduced in combination with flexible URS to improve UTUC detection [18]. In addition, flexible URS provides clinicians with a means of performing endoscopic treatment for low-risk UTUC, raising new concerns for the selection of eligible patients for kidney-sparing surgery. Apart from the advantages of flexible URS, research suggests that diagnostic URS, including flexible URS, has an adverse effect on cancer control. To date, such research has focused on the impact of diagnostic URS on IVR, regardless of the type of URS. In this retrospective study, we assessed the impact of diagnostic URS on IVR following RNU, according to the type of URS, either flexible or rigid.

## 2. Materials and Methods

### 2.1. Ethics

This study was performed in accordance with the tenets of the Declaration of Helsinki. The Institutional Review Board of Gangnam Severance Hospital approved the study protocol (approval number: 3-2020-0199, approval date: 2 July 2020).

### 2.2. Study Design and Population

This multicenter retrospective study involved two tertiary hospitals in the Republic of Korea. Medical records of 491 consecutive patients who underwent RNU between 2016 and 2019 were reviewed. Patients whose pathological diagnosis was not urothelial carcinoma were excluded. Other exclusion criteria were as follows: patients who underwent radical cystectomy prior to RNU or simultaneously with RNU, those who experienced IVR within 90 days after RNU, and patients whose follow-up period was <90 days. Finally, 396 patients were enrolled for analysis, with a median follow-up of 30 months (interquartile range [IQR]: 19–43).

### 2.3. Operation Techniques and Postoperative Management

A retrograde pyelography was performed with a 5-Fr ureteral catheter and diluted contrast media. Then, a guide wire was placed under visual and fluoroscopic control. The type of URS was selected at the surgeon’s discretion. For rigid URS, gravity-based irrigation was used; the irrigation fluid was placed 60 to 80 cm above the patient. For flexible URS, a ureteral access sheath was introduced along the guide wire below the tumor, and selective ureteral cytology was collected. Irrigation was provided with normal saline and a pressurized pump to ensure clear vision.

RNU was performed via open, laparoscopic, or robotic surgery in compliance with oncological principles. Bladder cuff resection was performed in all patients using an extravesical technique, and the intramural portion of the ureter was completely dissected. Lymph node dissection was performed at the surgeon’s discretion. No postoperative dose of intravesical chemotherapy, such as mitomycin, was administered. Adjuvant chemotherapy was administered to patients who had adverse pathological features such as pT3 or higher stage, lymph node involvement, lymphovascular invasion, or aggressive variant histology. All patients who underwent RNU were followed-up with abdomen-pelvis CT and chest CT every 3–6 months and with cystoscopy and urine cytology every 3 months.

### 2.4. Definitions of Groups, Outcomes, and Covariates

The study cohort was divided into three groups according to the type of diagnostic URS prior to RNU: non-URS, rigid URS, or flexible URS. The study outcome was defined as pathologically proven IVR occurring within 1 year of RNU. Time-to-event was defined as the duration between the date of RNU and the first IVR. The tumor locations for UTUC were divided into the renal pelvis and ureter. Tumor grade and pathological T stages of UTUC were described according to the 2016 World Health Organization/International Society of Urologic Pathology consensus classification and the 2010 American Joint Committee on Cancer and the International Union for Cancer Control tumor–node–metastasis classification [19,20].

### 2.5. Statistical Analysis

Demographic and clinicopathological factors were described and compared according to the type of diagnostic URS. Differences in categorical variables between the three groups were compared using Pearson’s chi-square test or Fisher’s exact test. Continuous variables were compared using the Mann–Whitney U test and described in a non-parametric manner. IVR-free survival curves were generated using the Kaplan–Meier method and compared using the log-rank test. A log–log plot and interaction test confirmed that the proportionality assumption was satisfied. Univariable and multivariable Cox proportional hazard models were used to estimate the risk of IVR according to covariates. A multivariable model was generated using the enter method, including factors that were significant (*p* < 0.05) or neared significance (*p* < 0.1) in univariate analysis, as well as factors known to affect IVR. Significance was set at *p* < 0.05, and all statistical tests were two-sided. All study analyses were performed using SAS^®^ System for Windows^®^ (version 9.4; SAS Institute Inc., Cary, NC, USA).

## 3. Results

In our study cohort, rigid and flexible URS was performed in 178 (45%) and 111 (28%) patients, respectively, and 107 (27%) patients did not undergo diagnostic URS (Table 1). Table 2 summarizes the comparisons of clinicopathological features according to the type of diagnostic URS. Significant differences in sex, urine cytology, tumor location, tumor grade, previous or concurrent bladder cancer, and smoking history were observed among the groups.

IVR was identified in 99 (25%) patients, specifically in 21, 41, and 37 patients in the non-URS, rigid URS, and flexible URS groups, respectively (Table 2). The univariable Cox regression analysis demonstrated that the flexible URS group had a higher IVR rate than the non-URS group (hazard ratio [HR] = 1.866; 95% confidence interval [CI], 1.092–3.188; *p* = 0.0224), with no significant difference between the rigid URS and non-URS groups (HR = 1.161; 95% CI, 0.686–1.966, *p* = 0.577) (Table 3 and Figure 1).

Multivariable Cox regression analysis also revealed that flexible URS was independently associated with an increased risk of IVR (HR = 1.807; 95% CI, 1.023–3.192; *p* = 0.0416), although rigid URS did not increase the risk of IVR (HR = 1.301; 95% CI, 0.759–2.230; *p* = 0.3388). A history of chronic kidney disease was also an independent predictor for an increased risk of IVR (HR = 1.795; 95% CI, 1.043–3.092; *p* = 0.0348). Adjuvant systemic chemotherapy was associated with a decreased risk of IVR (HR = 0.603; 95% CI, 0.368–0.989; *p* = 0.045) (Table 3). Carcinoma in situ, smoking history, and a history of hypertension showed some significance in the univariable analysis, but failed to reach statistical significance in the multivariable analysis.

## 4. Discussion

The impact of diagnostic URS on IVR remains controversial. Ishikawa et al. first described the impact of diagnostic URS on IVR, and several related studies have been performed [21]. Ishikawa et al. concluded that diagnostic URS has no adverse effects on IVR, and subsequent studies also drew a similar conclusion [16,21]. On the other hand, several studies did report adverse effects for diagnostic URS on IVR [17,22,23,24]. A recent meta-analysis consistently demonstrated an adverse effect of diagnostic URS on IVR [6,7,8]. Furthermore, differences according to the timing of diagnostic URS (simultaneously with RNU or before RNU) or whether the accompanied ureteroscopic biopsy was performed were also reported [25,26]. The mechanism through which diagnostic URS accelerates IVR has been explained by intraluminal seeding and cancer cell implantation promoted by either massive irrigation or ureteroscopic biopsy.

With the widespread use of flexible diagnostic URS, flexible URS allows urologists to examine all collecting systems, including the lower calyx, which cannot be visualized using rigid URS. Flexible URS has a smaller working channel than rigid URS, which results in a weaker irrigation capability. To maintain a better view for diagnostic purposes, a higher-pressure inflow of irrigation fluid is required, compared to that of rigid URS. Owing to these mechanical differences, flexible URS and rigid URS might pose different risks of IVR. To the best of our knowledge, this is the first study to investigate differences in the impact of diagnostic URS on IVR according to the type of URS, either rigid or flexible. Herein, multivariable analysis revealed that flexible URS was significantly associated with an increased risk of IVR, compared to non-URS, however, rigid URS was not. We conducted propensity score matching to adjust for demographic heterogeneity among three groups; it also supported these observations (Appendix A). Therefore, diagnostic URS should be performed as a guideline only when imaging and cytology are insufficient for the diagnosis of UTUC and/or for risk stratification of a tumor [4]. In particular, surgeons should consider that the risk of IVR might be increased when flexible diagnostic URS is indicated.

The potential mechanisms by which flexible URS increases the risk of IVR can be explained as follows. First, a high-pressure inflow of irrigation fluid is required to secure a clear field of view, owing to their smaller working channel, compared to rigid URS. To maintain a sufficient amount of irrigation fluid, the height of the irrigation fluid should be higher than usual or an irrigation pump is needed. Second, due to the smaller size of forceps used for ureteroscopic biopsy, considerably greater manipulation of cancer cells might occur to obtain a sufficient amount of tissue [27]. Third, an access sheath should be introduced to keep the intrarenal pressure low while performing flexible URS [28,29], resulting in increased irrigation fluid running through the collecting system. Therefore, cancer cells are likely to be exfoliated during flexible URS. From this point of view, the difference between rigid and flexible URS observed in this study may be due to a difference in the irrigation method rather than the difference in the instrument itself. These hypothetical explanations should be proven in clinical and preclinical studies.

Increased intrarenal pressure causes pyelovenous and pyelolymphatic backflow, which is theorized as a potential source of cancer cell transfer [8]. A recent literature review suggested that intrarenal pressure should remain <30 cm H_2_O during endoscopic procedures. However, the actual oncological risk promoted by diagnostic endoscopic procedures is uncertain due to the lack of strong evidence in real intrarenal pressure threshold [28]. Results of subsequent studies suggest that URS may not have a negative effect on cancer-specific survival or overall survival [17]. As aforementioned, although mechanical differences exist between flexible URS and rigid URS, previous studies do not distinguish the type of URS as either flexible or rigid. Since this study included relatively recent patients with UTUC to focus on IVR according to the type of URS, the follow-up period was insufficient to estimate either cancer-specific survival or overall survival. A longer follow-up period is required to estimate the effect of flexible URS on oncological outcomes other than IVR.

This study has several advantages in revealing the occurrence of IVR in patients who underwent RNU. Since most previous studies on IVR enrolled patients who underwent RNU for a period close to a decade, several confounding factors, such as surgeon factors and surgical modality, may have inevitably affected the analysis. However, this study was able to reduce heterogeneity because of the relatively short and recent period of study; thus, our study can reflect the current trends in real-world clinical practice. We attempted to minimize the possibility of newly developed primary bladder cancer independent of UTUC by limiting IVR to 1 year within RNU. Therefore, we focused on URS-induced IVR. In the period of this study, the two institutions rarely conducted neoadjuvant chemotherapy even in patients with T3 or higher clinical stage. Additionally, all bladder cuff management procedures were performed using an extravesical approach, and no postoperative intravesical instillation was noted, despite this being a known prevention therapy for IVR [30]. While this hindered us from estimating the effects of bladder cuff management, postoperative intravesical instillation, and neoadjuvant chemotherapy, it did allow us to concentrate on the effect of the type of URS on IVR.

Although we attempted to report IVR according to the type of diagnostic URS, this study has a few limitations. Owing to the retrospective design of this study, the absence of randomization resulted in heterogeneity among the three groups. A randomized clinical trial is needed to obtain definite conclusions on this issue, although it is rarely conducted for practical and ethical reasons. Moreover, we did not have objective data on the duration of the diagnostic URS procedure and the amount of irrigation fluid used during the procedure. Therefore, further research using more constructed data with a larger patient population is needed to account for differences in such characteristics among the study groups according to the type of URS.

## 5. Conclusions

In patients with UTUC, rigid URS may not increase the risk of IVR following RNU; however, flexible URS appears to be associated with a higher risk of IVR. When the diagnostic URS is inevitably implemented, physicians should consider that the risk of IVR may increase, especially when flexible URS is required. Our findings should be reproduced in large-population studies. Additionally, clinical and preclinical evidence supporting our results should be accumulated.

## Figures and Tables

**Figure 1 cancers-14-05629-f001:**
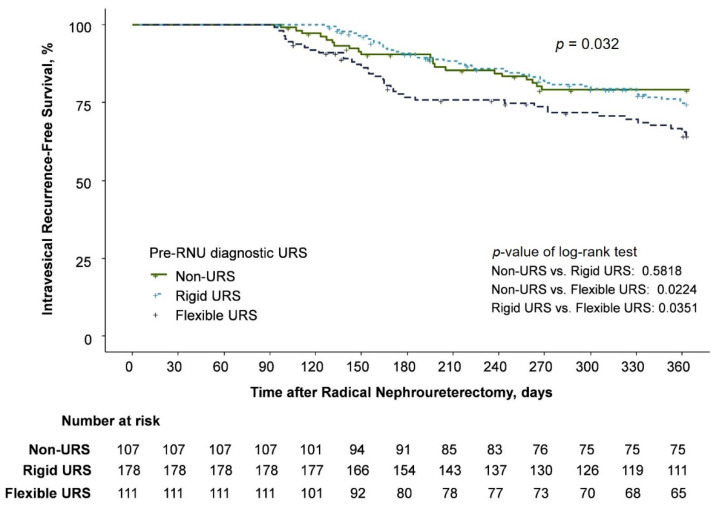
Kaplan–Meier estimates for intravesical recurrence-free survival.

**Table 1 cancers-14-05629-t001:** Clinicopathological characteristics of 396 patients with UTUC.

Age at RNU *, years	68.0 (60.0–76.0)
Sex	
Female	127 (32.07)
Male	269 (67.93)
HTN	236 (59.60)
DM	87 (21.97)
CAOD	46 (11.62)
COPD, asthma	35 (8.84)
CVA	26 (6.57)
CKD	45 (11.36)
Smoking history	
Never smoked	210 (53.03)
Ex- or current smoker	186 (46.97)
Tumor location	
Renal pelvis	220 (55.56)
Ureter	176 (44.44)
Laterality	
Left	198 (50.00)
Right	198 (50.00)
Previous/concurrent bladder cancer	81 (20.45)
Urine cytology	
Not done	136 (34.34)
Negative	140 (35.35)
Atypical cell	69 (17.43)
Positive	51 (12.88)
Type of URS	
Non-URS	107 (27.02)
Rigid URS	178 (44.95)
Flexible URS	111 (28.03)
Ureteroscopic biopsy	140 (35.35)
Surgical modality	
Open	105 (26.52)
Laparoscope or robot	291 (73.48)
Pathologic T stage	
pTa	35 (8.84)
pT1	126 (31.82)
pT2	80 (20.20)
pT3–4	155 (39.14)
Tumor grade	
Low grade	44 (11.11)
High grade	352 (88.89)
Pathologic N stage	
pN0	47 (11.87)
pNx	332 (83.84)
pN+	17 (4.29)
Tumor size	
<2 cm	66 (16.67)
≥2 cm	330 (83.33)
Lymphovascular invasion	89 (22.59)
Carcinoma in situ	55 (13.89)
Adjuvant chemotherapy	114 (28.79)

UTUC, upper tract urothelial carcinoma; URS, ureteroscopy; RNU, radical nephroureterectomy; HTN, hypertension; DM, diabetes mellitus; CAOD, coronary artery obstructive disease; COPD, chronic obstructive pulmonary disease; CVA, cerebrovascular accident; CKD, chronic kidney disease. Data are presented as *n* (%); * Age at RNU is presented as a median (interquartile range).

**Table 2 cancers-14-05629-t002:** Comparison of clinicopathological features according to the type of diagnostic URS.

	Non-URS Group	Rigid URS Group	Flexible URS Group	*p*-Value
No. of patients	107 (27%)	178 (45%)	111 (28%)	
Age at RNU *, years	69 (62–76)	68 (60–76)	68 (60–76)	0.633
Sex				<0.0001
Female	45 (42.06)	64 (35.96)	18 (16.22)	
Male	62 (57.94)	114 (64.04)	93 (83.78)	
HTN	58 (54.21)	111 (62.36)	67 (60.36)	0.3901
DM	25 (23.36)	30 (16.85)	32 (28.83)	0.0527
CAOD	6 (5.61)	26 (14.61)	14 (12.61)	0.0665
COPD, asthma	8 (7.48)	18 (10.11)	9 (8.11)	0.7124
CVA	8 (7.48)	13 (7.30)	5 (4.50)	0.5853
CKD	16 (14.95)	14 (7.87)	15 (13.51)	0.1326
Smoking history				0.0343
Never smoked	66 (61.68)	95 (53.37)	49 (44.14)	
Ex- or current smoker	41 (38.32)	83 (46.63)	62 (55.86)	
Tumor location				<0.0001
Renal pelvis	56 (52.34)	69 (38.76)	95 (85.59)	
Ureter	51 (47.66)	109 (61.24)	16 (14.41)	
Laterality				0.4409
Left	57 (53.27)	91 (51.12)	50 (45.05)	
Right	50 (46.73)	87 (48.88)	61 (54.95)	
Previous/concurrent bladder cancer	31 (28.97)	30 (16.85)	20 (18.02)	0.037
Urine cytology				0.0166
Not done	24 (22.43)	69 (38.76)	43 (38.74)	
Negative	41 (38.32)	56 (31.46)	43 (38.74)	
Atypical cell	17 (15.89)	33 (18.54)	19 (17.12)	
Positive	25 (23.36)	20 (11.24)	6 (5.40)	
Ureteroscopic biopsy	0 (0)	85 (47.75)	55 (49.55)	<0.0001
Surgical modality				0.4928
Open	31 (28.97)	42 (23.60)	32 (28.83)	
Laparoscope or robot	76 (71.03)	136 (76.40)	79 (71.17)	
Pathologic T stage				0.2172
pTa	10 (9.35)	13 (7.30)	12 (10.81)	
pT1	31 (28.97)	53 (29.78)	42 (37.84)	
pT2	25 (23.36)	42 (23.60)	13 (11.71)	
pT3–4	41 (38.32)	70 (39.33)	44 (39.64)	
Tumor grade				0.0224
Low grade	10 (9.35)	14 (7.87)	20 (18.02)	
High grade	97 (90.65)	164 (92.13)	91 (81.98)	
Tumor size				0.6782
<2 cm	15 (14.02)	32 (17.98)	19 (17.12)	
≥2 cm	92 (85.98)	146 (82.02)	92 (82.88)	
Lymphovascular invasion	23 (21.50)	41 (23.16)	25 (22.73)	0.9475
Carcinoma in situ	12 (11.21)	30 (16.85)	13 (11.71)	0.303
Adjuvant chemotherapy	26 (24.30)	60 (33.71)	28 (25.23)	0.1466
IVR within 1 year	21 (19.63)	41 (23.03)	37 (33.33)	0.0467

URS, ureteroscopy; RNU, radical nephroureterectomy; HTN, hypertension; DM, diabetes mellitus; CAOD, coronary artery obstructive disease, COPD, chronic obstructive pulmonary disease; CVA, cerebrovascular accident, CKD, chronic kidney disease; IVR, intravesical recurrence. Data are presented as *n* (%); * Age at RNU is presented as a median (interquartile range).

**Table 3 cancers-14-05629-t003:** Univariable and multivariable Cox regression analysis.

	Univariable Model		Multivariable Model	
	HR (95% CI)	*p*-Value	HR (95% CI)	*p*-Value
Type of URS				
Non-URS	Ref.		Ref.	
Rigid URS	1.161 (0.686–1.966)	0.577	1.301 (0.759–2.230)	0.3388
Flexible URS	1.866 (1.092–3.188)	0.0224	1.807 (1.023–3.192)	0.0416
Age at RNU	0.995 (0.976–1.015)	0.6428		
Sex				
Female	Ref.			
Male	1.150 (0.746–1.772)	0.5279		
HTN	1.428 (0.940–2.170)	0.0945	1.332 (0.869–2.041)	0.1884
DM	1.036 (0.650–1.652)	0.8813		
CAOD	1.536 (0.887–2.661)	0.1258		
COPD, Asthma	1.558 (0.810–2.999)	0.1841		
CVA	0.832 (0.364–1.898)	0.6613		
CKD	1.630 (0.954–2.784)	0.0737	1.795 (1.043–3.092)	0.0348
Smoking history				
Never smoked	Ref.		Ref.	
Ex- or current smoker	1.446 (0.974–2.148)	0.0676	1.390 (0.928–2.080)	0.1099
Tumor location				
Renal pelvis	Ref.			
Ureter	0.742 (0.494–1.115)	0.151		
Laterality				
Left	Ref.			
Right	0.909 (0.613–1.348)	0.6355		
Previous/concurrent bladder cancer	1.341 (0.847–2.124)	0.2105	1.170 (0.724–1.890)	0.5218
Urine cytology				
Negative	Ref.			
Atypical cell	0.959 (0.548–1.676)	0.8823		
Positive	1.378 (0.789–2.410)	0.26		
Ureteroscopic biopsy	1.133 (0.755–1.698)	0.5468		
Surgical modality				
Open	Ref.			
Laparoscope or Robot	0.985 (0.629–1.541)	0.9463		
Pathologic T stage				
pTa	Ref.			
pT1	1.391 (0.614–3.150)	0.4293		
pT2	1.747 (0.755–4.038)	0.1922		
pT3–4	1.277 (0.567–2.875)	0.5547		
Tumor grade				
Low grade	Ref.			
High grade	1.364 (0.688–2.707)	0.3743		
Tumor size				
<2 cm	Ref.			
≥2 cm	1.028 (0.602–1.755)	0.9202		
Lymphovascular invasion	0.731 (0.434–1.234)	0.2412		
Carcinoma in situ	1.582 (0.968–2.584)	0.0669	1.530 (0.926–2.527)	0.0969
Adjuvant chemotherapy	0.630 (0.389–1.021)	0.0605	0.603 (0.368–0.989)	0.0450

URS, ureteroscopy; RNU, radical nephroureterectomy; IVR, intravesical recurrence; HTN, hypertension; DM, diabetes mellitus; CAOD, coronary artery obstructive disease; COPD, chronic obstructive pulmonary disease; CVA, cerebrovascular accident; CKD, chronic kidney disease; DM, diabetes mellitus; CI, confidence interval; HR, hazard ratio; Ref., reference.

## Data Availability

The datasets generated during and/or analyzed during the current study are available from the corresponding author on reasonable request.

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
