# Peer review of "Intravesical Recurrence after Radical Nephroureterectomy in Patients with Upper Tract Urothelial Carcinoma Is Associated with Flexible Diagnostic Ureteroscopy, but Not with Rigid Diagnostic Ureteroscopy"

_cancers, 2022, doi:10.3390/cancers14225629_

Round 1

Reviewer 1 Report

Dear author

I would like to congratulate with you for the methodology and quality of the research and the statistics editing

As you remind in page 9 line 192 the use of UAS might be crucial for bladder seeding after flex-URS. Therefore I would create two subgroups in flex-URS group stratified by use of UAS and I would see if there are any difference among them and the other groups (semirigid-URS, non-URS)

Reviewer 2 Report

Post URS IVR is an important issue in the management of UTUC. This is a well-designed study. However, I would like to underline some parts;

-The technique of f-URS was not mentioned in the method part. f-URS can be performed over a guide-wire, through an UAS or as no-touch technique without any guide-wire or UAS. It is important to describe which and in what percent of these techniques are used, since they have different upper urinary tract manipulation and irrigation characters. This data should be add to the manuscript.

-Digital (re-use or single-use) flexible scopes have better image quality compared to fiberoptic scopes and this effects ureteroscopy time. I recommend to mention this issue in the discussion part.

-As stated by the authors, one of the main limitations of the study is the lack of information on ureteroscopy times and irrigation amounts.

Reviewer 3 Report

The purpose of the study was to analyze the impact of the ureteroscopic diagnosis on intravesical recurrence following radical nephroureterectomy for upper urothelial cell carcinoma, according to the type of ureteroscopy, rigid or flexible.

Even though the study idea is not new, the manuscript is well written, and every time new information is presented it is well received and regarded as auspicious.

The Material and Method describe a good, worked study design with nice data on the ethics, protocol, and statistical analysis. Also, the Results are supported by supplementary materials like tables and graphs.

Before publication, the article requires some minor modifications:

1.     I think the Introduction is too brief, and more than half of the references are cited here. More information about flexible ureteroscopy in the diagnosis of Upper Urinary Tract Urothelial Cell Carcinoma prior to radical nephroureterectomy could be added here.

2.     Please, check the language along the text

Reviewer 4 Report

This is an interesting study in that it is quite unexpected in the findings. The limits as you point are due to the retrospective review. This could open the door to further review.

Author Response

Answer) We deeply appreciate the reviewer’s comments.

Round 2

Reviewer 2 Report

I think the revised and final manuscript is acceptable for publication.